# Recognizing Post-Cardiac Injury Syndrome After Impella 5.5 Insertion in Cardiogenic Shock: A Case-Based Discussion

**DOI:** 10.3390/biomedicines13071737

**Published:** 2025-07-16

**Authors:** Aarti Desai, Shriya Sharma, Jose Ruiz, Juan Leoni, Anna Shapiro, Kevin Landolfo, Rohan Goswami

**Affiliations:** 1Division of Heart Failure and Transplant, Mayo Clinic, Jacksonville, FL 32224, USA; 2Department of Anesthesiology, Mayo Clinic, Jacksonville, FL 32224, USA; 3Department of Cardiovascular Surgery, Mayo Clinic, Jacksonville, FL 32224, USA

**Keywords:** pericarditis, PCIS, cardiogenic shock, Impella 5.5, heart transplant

## Abstract

The use of temporary mechanical circulatory support in refractory heart failure cardiogenic shock (HFCS) has risen, leading to potential complications. Post-Cardiac Injury Syndrome (PCIS) from Impella insertion is rare but may result from subclavian artery manipulation and aortic irritation. We report the first case of pericarditis (PCIS) caused by Impella 5.5 insertion in an HFCS patient awaiting heart transplantation. The patient developed chest pain, tachycardia, and hypotension post-Impella insertion. Laboratory results and electrocardiograms confirmed PCIS. Treatment with Ibuprofen and Colchicine was successful. He received a heart transplant 14 days later. This case emphasizes recognizing iatrogenic pericarditis after Impella insertion and the need to avoid additional myocardial strain.

## 1. Background

The use of mechanical circulatory support (MCS) devices is known to have the potential for vascular injury and systemic inflammatory response. Traditionally, this has been self-limiting; however, some case reports have suggested vascular inflammation that is invasive and can affect vessel integrity and lead to aortic valve and surrounding aortic tissue damage [1].

Post-implant complications of pericarditis, aortitis, and vascular inflammation may occur. Pericarditis, the most common pericardial disease, is an inflammation of the pericardium, the membrane enclosing the heart. It is a common cause of chest pain in young patients and is most commonly associated with viral infections. Rarely, pericarditis can occur after iatrogenic trauma and is known as Post-Cardiac Injury Syndrome (PCIS). Patients who develop PCIS present with signs and symptoms similar to those seen in patients with acute pericarditis, which includes a pleuritic chest pain, pericardial friction rub, ST-segment elevations, and a pericardial effusion [2].

The occurrence of PCIS in both adults and children has been documented to range from 10% to 40% following cardiac surgery. After the implantation of intra-cardiac devices, the reported incidence is approximately 1% to 5%. Following coronary intervention, the occurrence is estimated to be around 0.2% [3].

Most cases of pericarditis do not require specific treatment and can be managed with supportive care, such as non-steroidal anti-inflammatory drugs (NSAIDs) or colchicine, with rare refractory cases requiring corticosteroids. However, it is important to identify the underlying cause of pericarditis as some causes may have distinct prognostic and therapeutic implications [2].

Here, we report the first case of an Impella 5.5 implantation leading to systemic inflammation and evidence of pericardial/Post-Cardiac Injury Syndrome in a patient awaiting a heart transplant.

### 1.1. Impella 5.5 Device

The Impella 5.5 (Abiomed, Danvers, MA, USA) with Smart Assist is a temporary MCS (tMCS) that can be implanted through minimally invasive surgery, typically placed percutaneously or by surgical cutdown into the femoral or subclavian artery [4].

With flow rates of up to 6.2 L/min, the Impella 5.5 provides enhanced left ventricular (LV) unloading leading to improved hemodynamics and end-organ perfusion. Its utilization is growing in patients with acute decompensated heart failure, serving as a bridge to a durable LVAD, transplantation, or recovery [4]. In our case, we used the Impella 5.5 to augment left ventricular function while awaiting heart transplant. Unfortunately, in our patient, it led to the development of PCIS two days later.

### 1.2. Pathophysiology of PCIS Post-Impella Implantation

The exact pathogenesis of PCIS is not fully understood, but it is theorized to be an immune–inflammatory response triggered by vascular injury (Figure 1). The damage to mesothelial cells and the presence of blood in the pericardial space initiate an autoimmune reaction. This immune response is induced by cardiac antigens released during the injury, producing antibodies. The formed immune complexes are deposited in the pericardium, resulting in inflammation [5,6].

Although iatrogenic trauma during cardiovascular procedures, such as right heart catheterization, pacemaker lead insertion, percutaneous coronary intervention and radiofrequency ablation is commonly considered to be minimal, the convergence of this trauma, which may injure pericardial mesothelial cells and result in minor bleeding in the pericardial space, has the capacity to trigger inflammation and autoimmune responses in susceptible individuals and may lead to PCIS [6].

While the implantation of Impella 5.5 is considered to be a minimally invasive procedure, similar injury to myocardial cells and pericardium may occur leading to the development of PCIS (Figure 1). Diagnosis is further supported by ruling out other known causes of pericarditis such as viral infections, autoimmune conditions (systemic lupus erythematosus, arthritis), uremia, and malignancies.

## 2. Case Presentation

A 23-year-old African American male with a history of hypertension, non-ischemic dilated cardiomyopathy, morbid obesity, diabetes mellitus type 2, obstructive sleep apnea, atrial and ventricular arrhythmias, and prior pulmonary embolism, presented for evaluation for advanced medical therapies for heart failure. The patient underwent coronary angiography, which showed normal nonobstructive coronaries but a severely elevated left ventricular end-diastolic pressure at 30 mmHg. The echocardiogram revealed a severely reduced ejection fraction of around 20–25% with severe left ventricular dilation with a left ventricular end-diastolic dimension of 78 mm, and right ventricular systolic pressure 50 mmHg with moderate to severe tricuspid regurgitation, mitral regurgitation, and severe right ventricular failure.

The patient was experiencing symptoms of shortness of breath, dizziness, lightheadedness, significant bilateral lower extremity edema, and significant weight gain, despite recent hospitalization. He also complained of gastrointestinal discomfort and nausea. Further evaluation using a right heart catheterization (RHC) revealed an RA 6 mmHg, RV 34/8 mmHg, PA 39/25 mmHg (mean 30 mmHg), PCWP 30 mmHg, CO 3.78 L/min, CI 1.46 L/min/m^2^, and systemic vascular resistance (SVR) of 1502 dynes/cm5. Dobutamine 2.5 mcg/kg/min, milrinone 0.25 mcg/kg/min was initiated, and a single-chamber implantable cardioverter defibrillator was implanted for the primary prevention of sudden death due to arrhythmias. Guidelines directed medical therapy with apixaban 5 mg BID, Entresto 49–51 mg BID, spironolactone 25 mg daily, and dapagliflozin 10 mg daily was continued with a plan to discharge the patient with outpatient management. Unfortunately, over the next 18 days, the patient experienced worsening dyspnea, and hemodynamics deteriorated. Intermittent intravenous diuretic therapy with bumetanide 2.5 mg twice daily and chlorothiazide 500 mg twice daily was administered for volume management, and Dobutamine was increased to 5 mcg/kg/min. Repeat RHC showed BP 89/68 mmHg, RA 25 mmHg, PA 60/43 mmHg (mean 30 mmHg), PCWP 43 mmHg, CO 2.9 L/min, and CI 1.2 L/min/m^2^, and laboratory results showed lactate 3.1 mmol/L and SvO2 42%. Echocardiography showed no signs of pericardial effusion or thrombus and the LVEF was <10%. Given the worsening clinical status and impaired perfusion, a decision was made to implant Impella 5.5 to facilitate left ventricular unloading, pulmonary decongestion, and improve end-organ perfusion while he underwent heart transplant evaluation.

The Impella 5.5 was set to Performance Level (P level) P6 with a flow rate of 4.2 L/min, and a purge pressure of 506 mmHg was noted. The catheter position was assessed by transthoracic echocardiogram to accurately identify the cannula position relative to cardiac anatomic structures, which revealed the Impella device inlet 5.6 cm past the aortic annulus. The procedure was uneventful and did not require multiple position adjustments or catheter manipulation. The patient was systemically anticoagulated to prevent clot formation on the device with Bivalirudin 0.24 mg/kg/h.

The patient’s post-Impella hemodynamics were stable, but he experienced sharp chest pain localized to the left precordium within the first few hours after returning to the ICU. The pain was exacerbated by deep respirations and position changes. On examination, he was afebrile, HR 110 BPM, BP 110/70 mmHg, with MAP 83 mmHg. Auscultation revealed a new left parasternal pericardial friction rub. The electrocardiogram (ECG) showed widespread ST elevations and PR depression, indicative of pericarditis (Figure 2).

Echocardiography was performed to rule out pericardial effusion. The laboratory results revealed a C-reactive protein (CRP) level of 399 mg/L, Troponin T level of 27 ng/L, and erythrocyte sedimentation rate (ESR) of 75 mm/h. Considering clinical progression, examination findings, imaging and laboratory results, the diagnosis of Post-Cardiac Injury Pericarditis was made.

The patient received treatment with Ibuprofen 800 mg PO TID for two weeks and Colchicine 1.2 mg PO BID on the first day and 0.6 mg PO BID for another 10 days. The chest pain resolved within 3 days, and the patient resumed ambulation and daily activities without discomfort. Echocardiography performed 5 days later showed no evidence of pericardial effusion.

The patient remained hemodynamically stable, without the recurrence of pericarditis and successfully received a heart transplant 14 days after the development of PCIS.

## 3. Discussion

Complications such as stroke, pump thrombosis, hematoma formation, and device malfunction have been reported post-Impella implantation [7]. Here, we present the first-ever case of PCIS observed after Impella 5.5 insertion. We outline the diagnostic and management steps taken in this case to provide insight for other clinicians, as the use of tMCS in HFCS continues to increase.

### 3.1. Diagnosis and Management of PCIS

While PCIS does not have well-defined diagnostic criteria due to the lack of specific symptoms and signs, it should be considered after cardiac injury if at least two out of following five criteria are satisfied: unexplained fever, chest pain suggestive of pericarditis or pleurisy, audible pericardial or pleural rubs, evidence of pericardial or pleural effusion, and/or elevated CRP levels [5]. In our patient, three out of the five criteria were fulfilled (chest pain, elevated CRP, and pericardial friction rub) following Impella insertion leading the team to explore further using imaging.

Elevated white blood cells, ESR and CRP observed in our patient are seen in all cases of acute pericarditis irrespective of etiology. While these tests are non-specific, they suggest ongoing inflammation. Around 35% to 50% of patients exhibit elevated plasma troponin levels, which is also believed to be due to inflammation of the epicardium rather than myocardial necrosis [8].

Electrocardiography stands out as the most valuable diagnostic tool for acute pericarditis. It typically reveals a distinctive widespread upward concave ST-segment elevation, indicative of subepicardial inflammation [9]. This was seen in our patient. While the utility of echocardiography is restricted and limited in cases of uncomplicated acute pericarditis, it holds significance as a rapid, precise, and non-invasive method in evaluating potential consequences such as effusion or constriction which may lead to the development of constrictive pericarditis in the future [10,11]. We confirmed the absence of effusion using an echocardiogram in the post-Impella period.

Clinical trials propose different treatment regimens for the treatment of Pericarditis to reduce inflammation. Options include aspirin (e.g., 750 to 1000 mg every 8 h for 1 to 2 weeks with gradual tapering), ibuprofen (600 mg every 8 h as the initial dose), and indomethacin (50 mg every 8 h as the initial dose). Aspirin might be the preferred option for individuals with ischemic heart disease or those already using aspirin for other indications [11]. We treated our patient successfully using Ibuprofen (800 mg PO TID for two weeks) to help reduce inflammation and Colchicine (1.2 mg PO BID on the first day and 0.6 mg PO BID for another 10 days), an inhibitor of leukocyte migration and cytokine release known to reduce recurrence of pericarditis by 50%. For patients who fail to respond to, are unable to tolerate, or have medical contraindications to NSAIDs and colchicine, Corticosteroids (prednisone 0.2–0.5 mg/kg/day) may be used [9].

### 3.2. Impella Placement and Physical Therapy

While the management of pericarditis is generally conservative, as mentioned above, adequate Impella management is crucial to maintaining hemodynamic support in patients with cardiogenic shock. Our patient was awaiting heart transplantation, with the Impella used in the setting of progressive shock to optimize the patient’s clinical and hemodynamic profile. The Impella performance level was gradually increased over the PCIS treatment period from P6 to P8, increasing support from 4.2 to 5.1 L/min to allow enhanced LV unloading and reduce myocardial work. Inotrope support with Dobutamine 5 mcg/kg/min and milrinone 0.5 mcg/kg/min and systemic anticoagulation with Bivalirudin 0.05 mg/kg/h was continued as prior. Impella positioning in our patient was monitored with a weekly surface echocardiogram, and physical therapy continued without complications; however, careful securement using the three-point anchor system and our institutional-specific physical therapy protocol were used to avoid movement of the device within the axillary and aortic vessels (Figure 3) [12].

## 4. Limitations

Our case highlights a unique occurrence that may not be seen in low-volume centers. We have placed a total of 123 Impella 5.5 devices at the time of this case. We are limited in diagnostic data as a case, given the acute findings were halted in trend due to the patient receiving a life-saving organ transplant. Further safety data in multicenter trials may provide further insight.

## 5. Conclusions

Impella 5.5 is safe to use; however, rare complications may occur. Promptly identifying a probable etiology of Post-Cardiac Injury Syndrome holds significant therapeutic importance, as highlighted in our case. The typical scenario involving patients experiencing chest pain post-device placement should be assessed. The identification of PCIS after the insertion of Impella 5.5 was confirmed only after ruling out other prevalent causes, such as infections, autoimmune conditions, and malignancies.

## Figures and Tables

**Figure 1 biomedicines-13-01737-f001:**
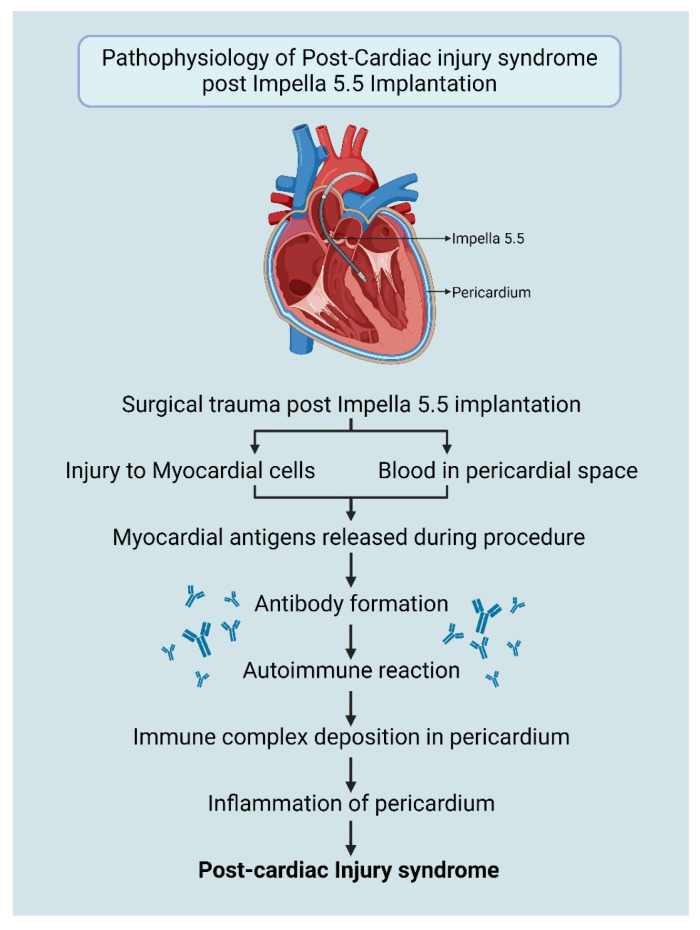
Pathophysiology of PCIS post-Impella 5.5 insertion. Created in BioRender. Aarti Desai. (2025). https://app.biorender.com/illustrations/6695149f2bf395e0bafd95e2.

**Figure 2 biomedicines-13-01737-f002:**
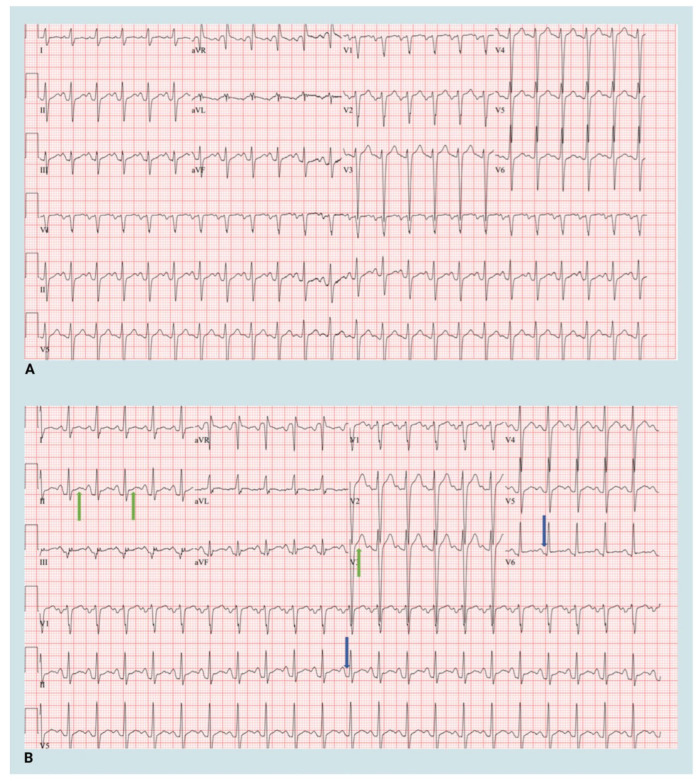
Electrocardiogram at baseline and post-Impella 5.5 implantation. (**A**) Baseline 12-lead electrocardiogram (**B**) 12-lead electrocardiogram post Impella 5.5 insertion. Green arrows—ST elevation, Blue arrows—PR depression. Created in BioRender. Aarti Desai. (2025). https://app.biorender.com/illustrations/66951dadb0681c95f4ec409d.

**Figure 3 biomedicines-13-01737-f003:**
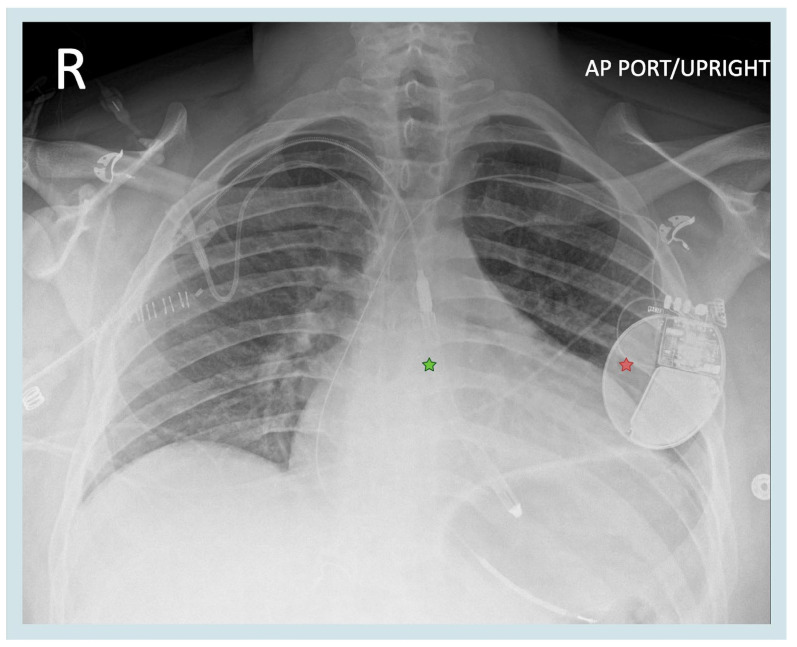
Position of the Impella 5.5 device. Green star—Impella 5.5. red star—implantable cardioverter defibrillator. Created in BioRender. Aarti Desai. (2025). https://app.biorender.com/illustrations/6698210d52e43397d44f1b26.

## Data Availability

The raw data supporting the conclusions of this article will be made available by the authors on request.

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
