# Peer review of "Recognizing Post-Cardiac Injury Syndrome After Impella 5.5 Insertion in Cardiogenic Shock: A Case-Based Discussion"

_biomedicines, 2025, doi:10.3390/biomedicines13071737_

Round 1
Reviewer 1 Report
Comments and Suggestions for Authors
- Low-voltage QRS complexes and electrical alternans are classic ECG features in pericardial disease. Were such changes observed or specifically excluded in this case? If present, how did they correlate with the outcome.
- Was the Impella 5.5 insertion procedure uneventful? Were there any potential precipitating factors such as multiple adjustments or repeated catheter punctures?
- Considering the patient has CKD stage 3, was there detailed monitoring of eGFR or Scr changes?
- The patient received continuous anticoagulation with Bivalirudin. Was the potential impact of this medication on the outcome considered?
Author Response
Dear reviewer,
Thank you for taking the time to review our case report. The revisions have significantly improved our manuscript.
- Low-voltage QRS complexes and electrical alternans are classic ECG features in pericardial disease. Were such changes observed or specifically excluded in this case? If present, how did they correlate with the outcome.
While low-voltage QRS and electrical alternans are indeed commonly associated with cardiac tamponade following pericardial inflammation, our patient fortunately did not progress to that stage most likely due to the timely and effective management of pericarditis provided by our team. - Was the Impella 5.5 insertion procedure uneventful? Were there any potential precipitating factors such as multiple adjustments or repeated catheter punctures?
It was uneventful. Added in lines 150-152 - Considering the patient has CKD stage 3, was there detailed monitoring of eGFR or Scr changes?
Thank you for your attention to detail. Upon reevaluation, we confirmed that the diagnosis of chronic kidney disease was included in error. We have re-checked the case, and the manuscript has been corrected. We sincerely regret the oversight. - The patient received continuous anticoagulation with Bivalirudin. Was the potential impact of this medication on the outcome considered?
Anticoagulation regimen post-impella implantation was maintained as per guidelines. No bleeding or thrombotic events were observed during Impella implantation or thereafter. Bival is not generally known to cause or aggravate pericardial diseases.
Additionally, we’ve made revisions to patient work-up lines 124-144 to clarify the timeline and reasoning behind management.
Sincerely,
Authors

Reviewer 2 Report
Comments and Suggestions for Authors
- The number of cases reported in this study is too limited.
- In addition, the key characteristics and aspects of the case should be clearly discussed and thoroughly reported.
- The period of study and observation is too short.
Author Response
Dear reviewer,
Thank you for taking the time to review our case report. We hope we have clarified the points mentioned below.
- The number of cases reported in this study is too limited.
This is a case report and to our knowledge, this is the first reported case of PCIS post-Impella 5.5 implantation. We have encountered only one such case at our institution, and no prior cases have been reported in the literature to date. - In addition, the key characteristics and aspects of the case should be clearly discussed and thoroughly reported.
The background section outlines the pathophysiology of PCIS development following Impella 5.5 implantation and the discussion is sectioned into the diagnosis and management of both PCIS and the Impella 5.5 device, with comparisons drawn between our case and existing guidelines point by point. - The period of study and observation is too short.
We present our patient’s clinical course beginning with admission for cardiogenic shock, followed by Impella 5.5 implantation and the subsequent development of PCIS. We describe our prompt diagnostic approach and successful management of PCIS. The case then concludes with successful heart transplantation.
Additionally, we’ve made revisions to patient work-up lines 124-144 to clarify the timeline and reasoning behind management.
Thank you for your time.
Sincerely,
Authors

Round 2
Reviewer 2 Report
Comments and Suggestions for Authors
The revised manuscript is acceptable.